# Neuroblastoma Molecular Risk-Stratification of DNA Copy Number and *ALK* Genotyping via Cell-Free Circulating Tumor DNA Profiling

**DOI:** 10.3390/cancers13133365

**Published:** 2021-07-05

**Authors:** Smadar Kahana-Edwin, Lucy E. Cain, Geoffrey McCowage, Artur Darmanian, Dale Wright, Anna Mullins, Federica Saletta, Jonathan Karpelowsky

**Affiliations:** 1Advanced Molecular Diagnostics, Children’s Cancer Research Unit, Kids Research, The Children’s Hospital at Westmead, Sydney, NSW 2145, Australia; FSaletta@ccia.org.au (F.S.); jonathan.karpelowsky@sydney.edu.au (J.K.); 2Cancer Centre for Children, The Children’s Hospital at Westmead, Sydney, NSW 2145, Australia; lucy.cain@health.nsw.gov.au (L.E.C.); geoff.mccowage@health.nsw.gov.au (G.M.); Anna.Mullins@health.nsw.gov.au (A.M.); 3Cytogenetics Department, Sydney Genome Diagnostics, The Children’s Hospital at Westmead, Sydney, NSW 2145, Australia; artur.darmanian@health.nsw.gov.au (A.D.); dale.wright@health.nsw.gov.au (D.W.); 4Division of Child and Adolescent Health, The University of Sydney, Sydney, NSW 2145, Australia; 5Paediatric Oncology and Thoracic Surgery, The Children’s Hospital at Westmead, Sydney, NSW 2145, Australia

**Keywords:** ctDNA, neuroblastoma, risk-stratification, *ALK* variants, treatment monitoring, pre-analytical

## Abstract

**Simple Summary:**

Neuroblastoma is composed of several molecular subtypes that are associated with distinct outcomes. Risk classification into these subtypes is routinely assessed at diagnosis and is based on the evaluation of well-defined chromosomal changes found in the tumor DNA which requires a tissue biopsy. However, current clinical practice limits the amount of tumor tissue available for molecular diagnostics. Tumors often shed cells and subcellular components into body fluids. Liquid biopsy is an emerging method to detect such cancer biomolecules in the bloodstream and holds great promise for personalized medicine. Here, we present the use of liquid biopsy to provide molecular classification for neuroblastoma that is independent to tissue biopsy and can potentially be used as a companion diagnostic test to reduce tissue requirements. We will also discuss the pre-analytical considerations for the successful application of this approach in the clinical setting.

**Abstract:**

Background: *MYCN* amplification (MNA), segmental chromosomal aberrations (SCA) and *ALK* activating mutations are biomarkers for risk-group stratification and for targeted therapeutics for neuroblastoma, both of which are currently assessed on tissue biopsy. Increase in demand for tumor genetic testing for neuroblastoma diagnosis is posing a challenge to current practice, as the small size of the core needle biopsies obtained are required for multiple molecular tests. We evaluated the utility of detecting these biomarkers in the circulation. Methods: Various pre-analytical conditions tested to optimize circulating-tumor DNA (ctDNA) copy number changes evaluations. Plasma samples from 10 patients diagnosed with neuroblastoma assessed for SCA and MNA using single nucleotide polymorphism (SNP) array approach currently used for neuroblastoma diagnosis, with MNA status assessed independently using digital-droplet PCR (ddPCR). Three patients (one in common with the previous 10) tested for *ALK* activating mutations p.F1174L and p.F1245I using ddPCR. Results: Copy number detection is highly affected by physical perturbations of the blood sample (mimicking suboptimal sample shipment), which could be overcome using specialized preservative collection tubes. Pre-analytical DNA repair procedures on ctDNA before SNP chromosome microarray processing improved the lower limit of detection for SCA and MNA, defined as 20% and 10%, respectively. We detected SCA in 10/10 (100%) patients using SNP array, 7 of which also presented MNA. Circulating-free DNA (cfDNA) and matched tumor DNA profiles were generally identical. MNA was detected using ddPCR in 7/7 (100%) of MNA and 0/12 (0%) non-MNA cases. MNA and *ALK* mutation dynamic change was assessed in longitudinal samples from 4 and 3 patients (one patient with both), respectively, accurately reflected response to treatment in 6/6 (100%) and disease recurrence in 5/6 (83%) of cases. Samples taken prior to targeted treatment with the ALK inhibitor Lorlatinib and 6–8 weeks on treatment showed reduction/increase in *ALK* variants according to response to treatment. Conclusions: These results demonstrate the feasibility of ctDNA profiling for molecular risk-stratification, and treatment monitoring in a clinically relevant time frame and the potential to reduce fresh tissue requirements currently embedded in the management of neuroblastoma.

## 1. Introduction

Neuroblastoma is the most common extracranial solid tumor in children, accounting for about 8% of all childhood cancers, and is diagnosed using a combination of laboratory tests, radiologic imaging and pathology assessment. Elevated urine catecholamine metabolites and imaging findings of meta-iodo-benzyl-guanidine (MIBG) radiotracer avidity (occurring each in approximately 90% of patients) are tested early in the diagnostic process and are sufficient for provisional diagnosis. On the other hand, histological classification, *MYCN* amplification (MNA), segmental chromosome aberrations (SCA) and *ALK* activating mutations detected via tissue biopsy, are the current molecular foundations of risk-group stratification and targeted therapeutics [1]. Increase in demand for tumor genetic testing for neuroblastoma diagnosis is posing a challenge to current practice, as the small size of the core needle biopsies obtained are required for multiple molecular tests: single nucleotide polymorphism (SNP) chromosome microarray analysis (CMA) or fluorescence in situ hybridization (FISH) for SCA and/or MNA, and DNA sequencing for *ALK* somatic variants [2].

Sampling body fluids such as peripheral blood, also known as “liquid biopsy”, offers an alternative biological source material for analysis of tumor-specific biomarkers within cell free circulating tumor DNA (ctDNA). Liquid biopsy is rapidly emerging as a revolutionizing approach for cancer diagnosis and treatment through interrogation of tumor-shed biomolecules [3]. The unique challenges in the pediatric setting, and neuroblastoma diagnosis, make such a minimally invasive approach to disease diagnosis highly attractive.

Over the past decade, there has been growing interest in the utility of ctDNA in neuroblastoma. Early efforts toward the noninvasive identification of recurrent prognostic copy number aberrations showed high sensitivity and specificity for MNA tumors (91% and 98%, respectively) [4], but only low-to-moderate rates for detection of 17q-gain (51–59% and 71–94%, respectively) [5] and 11q-loss (26% and 100%, respectively) [6]. These different rates are likely due to the masking effect of healthy circulating-free DNA (cfDNA) on moderate copy number changes, or potentially these aberrations being present at sub-clonal fractions. More recently, advances in genomic technologies made it possible to perform genome-wide assessment of genomic abnormalities from cfDNA, thereby capturing all neuroblastoma subtypes with 68–100% overall sensitivity (Reviewed in [7,8,9,10]). Chromosomal microarray is a key tool used for diagnosis of SCA (and MNA) in patients with neuroblastoma. Importantly, Chicard and colleagues [11] reported the use of a microarray for cfDNA analysis, reaching a sensitivity of 23–75%, which was dependent on disease stage with accurate copy number detection highly compromised due to: (1) background cfDNA shed from normal blood cells, and (2) the highly degraded nature of ctDNA, which may have reflected collecting blood under suboptimal conditions and preforming subsequent analysis. This may further hinder the utility of liquid biopsy in clinical practice.

Lately, the efficacy of ALK inhibitors had shown potential in *ALK*-driven refractory or relapsed neuroblastoma [12] and is currently being investigated as upfront treatment for newly diagnosed patients with high-risk disease and confirmed *ALK* mutated tumors (COG ANBL1531, ClinicalTrails.gov identifier NCT03126916). However, access to adequate core needle tissue to determine *ALK* somatic variants is challenging, with only 10% of samples reported to be successfully genotyped (either mutated or wild type) (COG personal communications). This is disappointing as the prevalence of *ALK* somatic variants in this population has been reported to be relatively high (14%) [13]. *ALK* somatic variants identified in cfDNA using digital-droplet PCR (ddPCR) in a series of small-cohort studies [14,15] highlight the feasibility of liquid biopsy in this setting.

The aim of our study was to evaluate and optimize pre-analytical processes for detecting neuroblastoma biomarkers in the circulation. We studied a series of 10 cfDNA samples obtained from patients with neuroblastoma and assessed whether SCA and MNA could be identified using various modified protocol conditions. We also evaluated the dynamics of detectable MNA and *ALK* variants throughout the course of treatment of 6 patients, 3 of which treated with the ALK inhibitor Lorlatinib, to assess ctDNA utility monitor therapeutic biomarkers and treatment efficacy in real-time.

## 2. Materials and Methods

### 2.1. Patients and Samples

Eligibility criteria for this prospective observational study included diagnosis of neuroblastoma and confirmed MNA status. Accordingly, sixteen patients enrolled: 13 at initial diagnosis, and 3 at disease relapse. The study was conducted under research protocols approved by the Sydney Children’s Hospitals Network Human Research Ethics Committee, (reference number: HREC/17/SCHN/302). Forty-four samples (from thirteen patients) prospectively collected from patients treated at the Children’s Hospital at Westmead (CHW), with all patients’ parents consented to the study. Additional thirteen samples collected from children in long-term remission from solid cancers (*n* = 3 neuroblastoma, *n* = 5 hepatoblastoma, and *n* = 5 sarcomas) who are otherwise healthy. Additional two and one retrospective samples obtained from Children’s Cancer Institute Tumour Bank (CCI-TB) (NSW) and from Zero Childhood Cancer (ZCC) PRISM Program (NSW), respectively. Additional clinical, pathological, and radiological information is found in Appendix A.

Blood samples collected in Streck (Cell-Free DNA BCT^®^, STRECK, La Vista, NE, USA, catalog No. 218997) or EDTA tubes and processed for plasma. Samples from CHW/ZCC collected in Streck and processed at ambient temperature in a double-centrifugation protocol: first centrifugation at 1600× *g* for 10 min followed by plasma supernatant aspiration into new tubes without disturbing the buffy coat layer, then a second centrifugation of the plasma supernatant at 15,500× *g* for 10 min, followed by aspirating the top of phase into new tubes without disturbing the pellet, and storing at −80 °C until DNA isolation. Samples collected at CCI-TB (in EDTA) centrifuged at approximately 1000× *g* for 10 min before plasma supernatant isolation and storage at −80 °C. An additional high-speed centrifugation step (15,000× *g* for 10 min at 4 °C) after thawing of stored samples was preformed prior to DNA extraction.

For Section 3.2.2, blood samples collected in Streck, PAXgene (PAXgene^®^ Blood ccfDNA Tube, product reference 768165, BD Biosciences Macquarie Park, NSW, Australia), and EDTA tubes. Samples processed in a double-centrifugation protocol as described above, with Streck and PAXgene tubes processed at ambient temperature and samples in EDTA processed at cooled 4 °C conditions. Gentle agitation to mimic sample transfer conditions performed using horizontal orbital rotation at a low speed of 3.75 RPM.

### 2.2. DNA Isolation

Cell free circulating DNA was extracted from 0.25–3 mL of frozen plasma samples using the QIAamp Circulating Nucleic Acid kit (Qiagen, catalog No. 55114, Chadstone, VIC, Australia) according to the manufacturer’s instructions, except for increasing the proteinase digest step to 60 min for plasma samples collected in Streck and PAXgene tubes, as recommended by Streck and PAXgene tube product literature. In cases of starting volume of plasma lower than 3 mL, volumes adjusted to 3 mL with PBS. DNA was eluted in 50 µL buffer provided with the kit and stored at −80 °C until PCR analysis. DNA amount was evaluated using Qubit dsDNA High Sensitivity Assay Kit (Invitrogen™, Thermo Fisher Scientific, Waltham, MA, USA).

Genomic/constitutional DNA was extracted from cell lines and whole blood samples using AllPrep DNA/RNA/Protein kit (Qiagen; catalog No. 80004) and QIAGEN DNeasy Blood & Tissue kit (Qiagen; catalog No. 69504), respectively, according to the manufacturer’s instructions. Cell lines used in this study: neuroblastoma cell lines-Kelly, BE2C, IMR32, SK-N-SH purchased from ECACC (European Collection of Authenticated Cell Cultures, Salisbury, UK), and sarcoma cell lines–RH30, A673, and ES8 kindly provided by Dr Belinda Kramer (Advanced Cellular Therapeutics, Children’s Cancer Research Unit, Kid’s Research, The Children’s Hospital at Westmead, NSW, Australia). All cell lines subjected to Short Tandem Repeat (STR) profiling by CellBank Australia (Westmead, Australia) to confirm identity.

### 2.3. cfDNA Treatment

Purified cfDNA was treated with either with the Infinium^®^ HD FFPE DNA Restore Kit (Illumina, San Diego, CA, USA) or SureTag Purification Columns (Agilent, Santa Clara, CA, USA), according to the manufacturers’ instructions.

### 2.4. SNP Chromosomal Microarray Analysis (SNP CMA)

SNP CMA was performed with the Infinium CytoSNP−850K v1.2 Beadchip (Illumina, San Diego, CA, USA), according to the manufacturer’s instructions. Microarray data was analyzed using BlueFuse Multi v4.5 software (Illumina, San Diego, CA, USA) based on the reference human genome (hg19/GRCh37) according to laboratory protocols within the CHW Cytogenetics Department.

Satisfactory SNP microarray data quality control (QC) metrics are usually defined by Median Log R Deviation (Log R Dev) and Median B-Allele Frequency Deviation (BAF Dev) and SNP call rate. For DNA samples with high molecular weight and high purity, satisfactory QC metrics are <0.2, <0.03 and >0.98, respectively. It was anticipated that SNP CMA QC metrics using ctDNA samples would not meet these typically values.

The mean effective resolution for detecting copy number abnormalities is ~20–50 kb and SCAs were defined as being >5 Mb in size. Smaller abnormalities were defined as focal gain or loss.

### 2.5. ddPCR

(1) MNA: We designed a ddPCR assay to evaluate *MYCN* copy number levels compared with the reference gene *THNSL2*. *THNSL2* was chosen as it is positioned near the centromere of the same chromosome arm as *MYCN*, to control for a false positive result due to numerical change of whole chromosome 2. We also included evaluation of ALK gene levels as: (a) *ALK* amplification is known to occur in 3–4% of neuroblastoma sporadic cases [16] almost exclusively in the context of MNA due to their close proximity on chromosome 2p23–24 [17], and (b) to increase the robustness for normalization by using both *ALK* and *THNSL2* as reference genes. Simultaneous detection of the three target genes using only two fluorophores included strategies to optimize primer/probe-sets concentration that allow a clean segregation of the droplets by levels of fluorescence in the data output [18] (Appendix A).

Samples analyzed for *MYCN*, *ALK*, and *THNSL2* genomic regions using PrimePCR Copy Number Assays: dHsaCP2506554 (*MYCN*, HEX), dHsaCP1000588 (*ALK*, 6-FAM), and dHsaCP2506297 (*THNSL2*, 6-FAM), respectively (Bio-Rad Laboratories, Gladesville, NSW, Australia). Duplex *MYCN*/*THNSL2* ddPCR reaction consisted of 10 μL ddPCR™ Supermix for Probes (No dUTP) (Bio-Rad Laboratories), 1 μL of *MYCN* and *THNSL2* PrimePCR assay (final concentrations of primers and probes of 900 nM and 250 nM, respectively), 4 units of HaeIII restriction enzyme (New England BioLabs, Notting Hill, VIC, Australia), and cfDNA/water for a final volume of 20 μL. For triplex *MYCN*/*ALK*/*THNSL2* ddPCR, additional 1 μL of *ALK* PrimePCR assay was included, while *THNSL2* PrimePCR assay was reduced to 0.4 μL. In duplex ALK/THNSL2 dHsaCP2506297 (*THNSL2*, HEX) was used. NTCs contained purified water instead of cfDNA. The ddPCR reaction mixture was used for droplet generation, and amplification was carried out in a C1000 Touch Thermal Cycler (Bio-Rad Laboratories) under the following conditions: 95 °C for 10 min, 40 cycles of 94 °C for 30 s, 60 °C for 1 min; then 98 °C for 10 min. ddPCR was performed using the QX200 ddPCR system according to manufacturer’s instructions (Bio-Rad Laboratories). QuantaSoft™ Analysis Pro v1.0 software (Bio-Rad Laboratories) used for data analysis. Target and reference/s copies were within the dynamic range of the instrument to ensure accurate detection level. Each sample was tested in 2 to 3 replicates preformed in at least 2 different experiments.

(2) ALK variants: NM_004304.5 (ALK):c.3522C > A (p.F1174L) [*ALK* p.F1174L] analyzed using PrimePCR Mutation Detection Assays: dHsaCP2000084 (p.F1174L c.3522C > A, HEX) and dHsaCP2000083 (*ALK* wild type, 6-FAM) as described above, with Tm 55 °C.

NM_004304.5 (ALK):c.3733T > A (p.F1245I) [*ALK* p.F1245I] analyzed using primers: 5′-TGGCTGTCAGTATTTGGAGG-3′ and 5′ CAGGAAGAGCACAGTCACTT 3′, wild type probe: 5′-[HEX] A+ACCACTTCATCCACCGGTGA-[IABkFQ]-3′, and *ALK* c.3733T > A probe: 5′-[6-FAM] AACCACATCATCCACCGGTGAGT [IABkFQ]-3′, as described above (A+-LNA nucleotide). Wild type and variant *ALK* sequences were used following the dMIQE guidelines [19].

*ALK* assays tested positive on genomic DNA from the primary tumors of Dx1, Dx9, and Rec3 to demonstrate VAF of 19.13%, 35.15%, and 29.51%, respectively. Tm optimizations conducted on tumor DNA. All cell lines used in this study tested negative for either p.F1174L or p.F1245I, with the exception of Kelly and SK-N-SH which tested positive for *ALK* p.F1174L, confirming the specificity of the assays.

### 2.6. qPCR

Samples analyzed for *MYCN*, and *THNSL2* genomic regions in the Rotor Gene Q (QIAGEN) real-time PCR, similarly to ddPCR: qPCR reactions consisted of 10 μL ddPCR™ Supermix for Probes (No dUTP) (Bio-Rad Laboratories), 0.8 μL of *MYCN* and *THNSL2* PrimePCR Copy Number Assays (dHsaCP2506554, and dHsaCP2506297, respectively; final concentrations of primers and probes of 500 nM and 200 nM, respectively), 4 units of HaeIII restriction enzyme (New England BioLabs), and cfDNA/water for a final volume of 20 μL. NTCs contained purified water instead of cfDNA. Amplification was carried out in a Rotor Gene Q (QIAGEN) under the following conditions: 95 °C for 10 min, 40 cycles of 94 °C for 30 s, 60 °C for 1 min; then 98 °C for 10 min. Each sample was tested in 3 replicates. Data collected from the yellow (HEX) and green (FAM) channels. Rotor Gene Q Series Software 2.3.1 (QIAGEN) was used for data output, with thresholds adjusted manually to generate equal copy number with minimal variance of *MYCN* and *THNSL2* in cell lines samples A673 and ES8. Primers’ efficiencies calculated from serial dilution of Kelly cell line DNA generating matched *MYCN* and *THNSL2* calling in appropriate dynamic range to ensure accurate detection level. Pfaffl method [20] used to calculate *MYCN* copy number:(1)$$MYCN\;\mathrm{copy}\;\mathrm{number}\;=2\;\times \;\frac{MYCN\;{\mathrm{primers}\;\mathrm{efficiency}\;}^{MYCN\;\mathrm{Ct}}}{THNSL2\;{\mathrm{primers}\;\mathrm{efficiency}\;}^{THNSL2\;\mathrm{Ct}}}$$
95% CI calculated with t statistics for 3 degrees of freedom.

### 2.7. Statistical Analysis

Mann–Whitney nonparametric *t* test, paired *t* test, Pearson correlation test, and Bland-Altham method comparison test were performed using GraphPad Prism version 9.1.1 for macOS, GraphPad Software, San Diego, CA, USA, www.graphpad.com (accessed on 15 April 2021).

## 3. Results

### 3.1. SCA and MNA Detection by SNP CMA

#### 3.1.1. Pre-analytical Optimization of cfDNA

Plasma samples from 16 patients with neuroblastoma and 13 children in remission from various types of solid cancers were used for cfDNA extraction. In accordance with previous report [11] compared with cfDNA in healthy individuals, cfDNA concentrations in patients with neuroblastoma were measured at extremely high levels with a wide range (21–9980 ng/mL of plasma, mean 1254 ng/mL, median 172 ng/mL; Mann–Whitney test, *p* < 0.0001), which was dependent on disease stage for treatment naïve samples (median 1008 ng/mL and mean 2426 ng/mL for metastatic disease versus 42 ng/mL and 54 ng/mL for localized disease, respectively; Mann–Whitney test, *p =* 0.004) (Appendix A).

From these, 3 samples with cfDNA concentration >1500 ng/mL were used for assay optimization of the CytoSNP-850K Beadchip assay, where 150 ng cfDNA was manipulated as follows: (1) untreated, (2) enriched for DNA fragments >150 base pairs using. SureTag columns, and (3) underwent DNA repair using the FFPE DNA Restoration kit.

Enrichment for large cfDNA fragments using the SureTag columns did not significantly improve microarray data QC metrics: Log R Dev reduced from mean 0.57 to 0.56) (Appendix A). However, treatment with the FFPE DNA Restoration kit reduced the noise compared with the untreated cfDNA and provided tight profiles, enabling clear identification of SCA (Log R Dev reduced to mean 0.43, paired *t*-test *p <* 0.05) (Figure 1). Although, Log R Dev was not optimal (see methods above), the BAF and Log R ratio (copy number) plots showed improvement with SCA and MNA more easily identifiable following FFPE DNA restoration treatment.

#### 3.1.2. cfDNA SNP CMA Limit of Detection (LOD)

High-density SNP microarray can detect copy number abnormalities (CNA) in tissue at clonal levels >10–20% of the sample [21,22]. In order to evaluate the CytoSNP-850K Beadchip assay sensitivity to detect CNAs in cfDNA, we serially titrated cfDNA into matched constitutional DNA from one patient (ID: Dx1) to produce artificial sub-clonal fractions of CNAs at levels of 0%, 10%, 20%, 30%, 40%, 50% and 100% (Constitutional DNA was sheared to generate fragmented DNA of similar size distribution as cfDNA (Appendix A) [23]. Both cfDNA and sheared constitutional DNA were treated with the FFPE DNA Restoration kit prior to titration. Subsequently, a total of 150 ng DNA of each titrated level was then used as starting material for the 850 CytoSNP Beadchip assay. Despite the suboptimal QC metrics (see above Log R Dev result), an equivalent assay sensitivity or LOD for SCA (LOD 20%) and MNA (LOD 10%) was found between ctDNA and high molecular weight tumor DNA (Figure 2).

#### 3.1.3. cfDNA SNP CMA Evaluation of 10 Cases with Neuroblastoma

Since SCA and MNA were shown to be detectable using 150 ng cfDNA treated with the FFPE DNA Restoration kit, we further challenged the lower sample input limit using 10 patients in our cohort with available cfDNA greater than 30 ng (range: 30–150 ng, mean 97 ng). None of the SNP array runs failed. 10/10 (100%) patients showed SCA: 8/8 (100%) at initial presentation with metastatic disease, and 2/2 (100%) at metastatic relapse. Of these, which 7 also showed MNA. cfDNA and matched tumor DNA profiles were generally identical in 9 patients, however, MNA without SCA was identified in one patient tested at relapse (Appendix A).

### 3.2. MNA Detection via ddPCR

#### 3.2.1. MNA-ddPCR Assay Design

ddPCR is a highly sensitive technology to quantify specific gene regions from low input samples, including cfDNA from neuroblastoma samples [15,16]. We designed a ddPCR assay to evaluate *MYCN* copy number. In order to reduce cfDNA input requirements, we designed a triple amplitude-based multiplex (triplex) ddPCR assay that reliably quantified *MYCN*, *ALK*, and *THNSL2* copy number levels simultaneous. Assay specificity, dynamic range, and thresholds for gain and amplification were evaluated in MNA neuroblastoma cell lines [Kelly, BE2C, IMR32], a non-MNA neuroblastoma cell line [SK-N-SH], and non-MNA sarcoma cell lines [RH30, A673, and ES8], to demonstrate MNA levels ranging from 66–1038 in MNA-cell lines, *MYCN* gains of 2.9 and 3.4 copies in SK-N-SH and RH30, respectively, and normal diploid *MYCN* copy levels of 2.0 in A673 and ES8. *ALK* gains of 7.2, 4.4, 3.1, and 3.0 copies, were detected in cell lines Kelly, BE2C, SK-N-SH, and RH30, respectively (Appendix A). The accuracy and sensitivity of the triplex ddPCR assay was compared to those of duplex *MYCN*/*THNSL2* and *ALK*/*THNSL2* reactions in a subset of the cell lines, demonstrating 100% concordance (Appendix A). The LOD was evaluated in titration experiments by diluting BE2C (harboring MNA level of 530 copies) into ES8. MNA was detected at levels of 10% and *MYCN* gain at 0.5% BE2C, with a threshold of 2.5 for copy number changes (Appendix A).

We also compared results between ddPCR and qPCR for MNA detection, using duplex qPCR of *MYCN* and *THNSL2*. We found an excellent concordance between the two platforms for detecting normal copy levels, *MYCN* gain, and MNA (Pearson correlation coefficient r = 0.9996 (0.9965–1.000, 95% CI), *p* < 0.0001). However, the Poisson distribution and fitting algorithm used in ddPCR calculated higher levels with lower variance of MNA and *MYCN* gain compared with the Pfaffle method (Ref) and t statistics used in qPCR, suggesting a superior sensitivity for ddPCR to detect copy number changes (Bland–Altman plot with Bias of −24.35, Appendix A).

#### 3.2.2. Pre-analytical Considerations for Optimized MNA ddPCR Detection

Published studies comparing different blood collection tubes demonstrated that testing for point mutations in ctDNA is optimally performed on plasma collected in DNA and cell stabilization tubes (e.g., Streck, PAXgene) or EDTA tubes, provided EDTA specimens are processed within 6 h of collection and stored at 4 °C (reviewed in [17]). However, the pre-analytical conditions suitable for copy number detection require further investigation. While point mutation events could be specifically detected, copy number detection is highly dependent on the amount of normal background cfDNA and might be missed under suboptimal collection and transport conditions. For example, Lodrini and collogues [16] reported lower *MYCN* copy numbers in cfDNA than matched tumor DNA which could be attributed to dilution effect from damaged white blood cells arising from suboptimal pre-analytic handling of the blood samples. 

As a proof of principle, we evaluated the detectability of MNA in different blood collection tubes [Streck, PAXgene, and EDTA], and under different conditions [plasma processed immediately or after 24 h incubation at room temperature, incubated either static or with a gentle agitation to mimic sample transfer conditions]. We spiked 4.5 ng Kelly DNA (representing about 750 cells) into 1 mL aliquots of blood from a healthy individual and plasma processed at room temperature (Streck and PAXgene), or at 4 °C (EDTA), accordingly.

*MYCN* copy number assessed using duplex ddPCR (Figure 3A). Generally, *THNSL2* concentration remained constant at each time-point across all blood collection tubes, with 1.6- to 2.5-fold increase observed in samples with delayed processing. The only exception was EDTA samples processed after 24 h of agitation, for which 16-fold increase was recorded and resulted in normal diploid *MYCN* levels. In addition, a reduction in MNA levels was noted after 24 h in all blood collection tubes that was partially due to increased *THNSL2* levels, but also to decreased *MYCN* concentration. The decrease in *MYCN* concentration was potentially caused by Kelly DNA degradation, while *THNSL2* cfDNA was not affected likely due to the protection within the nucleosomes [18]. MNA was highest in EDTA samples but also the most unreliable. Streck or PAXgene tubes performed equally with or without physical perturbations. Streck tubes demonstrated a slight advantage in maintaining MNA across all time-points and conditions compared with PAXgene tubes.

The effect of blood collection tube on MNA was also demonstrated in samples from a patient diagnosed with MNA-neuroblastoma (Figure 3B). Blood samples collected in EDTA and Streck from two patients at two clinical milestones, were split and processed for plasma at different time points after being left still at room temperature. For patient ID Dx3 we found no difference in the ability to detect MNA for either treatment naïve samples taken at diagnosis or samples taken after one cycle of chemotherapy. Samples obtained in EDTA and processed after 4 h or 24 h incubation at room temperature showed no disadvantage in detecting MNA. On the contrary, patient ID Dx1 had significant differences in the ability to detect MNA for either samples taken at disease recurrence or after one cycle of chemotherapy. MNA was detected only in plasma samples collected in Streck or EDTA tubes processed 1.5 h post blood draw. Serial matched samples collected in EDTA processed 5 h onwards were able to indicate only mild *MYCN* gain at decreasing levels throughout the timeline. In agreement, *THNSL2* reference gene concentration was lowest in the samples collected in the Streck or EDTA tubes processed within 1.5 h but increased to over 3–53-fold in the EDTA tubes processed at later time points. Although a larger set of matched samples needs to be examined to confirm these results, they underscore the problem of assessing copy number abnormalities collected under suboptimal conditions.

#### 3.2.3. MNA ddPCR Detection in Plasma Samples from Patients with Neuroblastoma

*MYCN* and *ALK* copy number was assessed in cfDNA samples from 16 patients with active neuroblastoma (14 at initial presentation and 2 at metastatic recurrence) and 3 patients in remission (Figure 4). MNA was detected only in 7/7 (100%, 5 at initial presentation, and 2 at recurrence) active MNA-neuroblastoma at 83–375 copies. *ALK* gains of 3.1–4.2 also detected in 3 of those 7 cases (43%). *MYCN* and *ALK* gains of 3.7 and 2.9, respectively, were detected in a patient with 4 and 3 copies, respectively, as found in matched diagnostic tumor DNA by SNP array analysis. The remaining 9 samples from active non-MNA patients and samples from 3 patients in remission showed normal diploid levels of both *MYCN* and *ALK*.

### 3.3. ctDNA Dynamics as an Indicator of Treatment Response

We took a longitudinal approach to molecularly assess treatment response via liquid biopsy, collecting multiple plasma samples throughout the course of treatment of 6 patients (2–8 samples from each): 4 patients with MNA neuroblastoma, and 3 patients with *ALK* variants in their tumor sample (2 with p.F1245I, and 1 with p.F1174L). One patient carried co-occurring MNA and *ALK* F1245I.

MNA and *ALK* variants were evaluated by ddPCR and correlated with clinical milestones and MIBG avidity (Figure 5). MNA and/or *ALK* variants detected in treatment naïve samples taken at initial diagnosis in 3/3 (100%) patients were completely abolished post neoadjuvant chemotherapy of 4–5 cycles, even though radiological studies still demonstrated MIBG avidity (though reduced). MNA and/or *ALK* variants were evident at disease recurrence of 5/5 (100%) patients. This data suggests a positive correlation between ctDNA and tumor burden.

Samples from the patient with the co-occurring MNA and *ALK* p.F1245I showed similar dynamics for both analytes, except for one sample taken during neoadjuvant chemotherapy in which *MYCN* gain of 6 copies was detected while *ALK* F1245I not identified, demonstrating that MNA is more readily detected in low ctDNA concentrations than single nucleotide variants (SNV) [or point mutation], in accordance with our results for better MNA detection over SCA using SNP CMA.

Three patients with *ALK* variants were treated with the ALK inhibitor Lorlatinib upon disease progression. All patients eventually developed resistance to treatment and relapsed on treatment. Nevertheless, *ALK* variants were assessed prior to and 6–8 weeks during Lorlatinib monotherapy and correlated with response to treatment (Figure 6): (1) a significant reduction in *ALK* variant to 2% for a patient with a prolonged response to treatment (12 months, Dx9), (2) 69% reduction in *ALK* variant in a patient with partial response to treatment (4 months, Rec3), (3) 5-fold increase in *ALK* variant in a patient with progressive disease (Dx1).

## 4. Discussion

Over the past decade, there has been growing interest in the utility of circulating biomarkers such as ctDNA and liquid biopsy of cancers. This approach can provide a global picture of the cancer in real time, at multiple time points, and with minimal invasiveness. A number of studies were able to accurately detect MNA and SCA in serum and plasma from patients with neuroblastoma as a tumor surrogate with varied diagnostic accuracies, which depended on disease stage [7]. In more recent years it became clear that mutations of the *ALK* tyrosine kinase domain constitute an important potential therapeutic target in neuroblastoma [12] and can be detected in ctDNA [14,15]. In the current clinical setting, surgical or core biopsies are carried out to provide histological diagnosis. However, these techniques do not always provide enough material for the analysis of biological markers. Here, we examined cfDNA isolated from plasma samples collected from patients with neuroblastoma and found that ctDNA is the predominant DNA type in plasma of metastatic patients and thus represents a relevant model for ctDNA analysis. We present strategies for optimizing plasma collection and ctDNA testing ensuring the accuracy and reproducibility of those analytes, which is required for clinical application of circulating biomarkers.

The combination of chromosomal microarray and FISH analyses is the current standard of practice for clinical genomic investigations for the detection of SCA and MNA, respectively, in neuroblastoma molecular risk-stratification. In the present study, we examined the utility of ctDNA to provide SCA and MNA profiles using SNP array, with orthogonal conformation of MNA status via ddPCR. SNP CMA is a key technique already incorporated into routine clinical diagnostic workflows. A major challenge associated with applying SNP CMA for ctDNA is providing adequate DNA input quantities which are not readily available from most cfDNA cancer samples. Moreover, neuroblastoma is diagnosed mostly in infants and very young children (90% younger than 5 years, median age 18 months) [1] which therefore able to provide only limited amounts of blood for a liquid biopsy test (typically 0.5–1 mL plasma). Hence, 200 ng input DNA required according to manufacturer’s recommendation for the Infinium CytoSNP-850K Beadchip array is not feasible for many patients with neuroblastoma, including all the patients in our cohort with localized disease. Here, the Infinium^®^ FFPE DNA Restoration kit has been employed for cfDNA repair and increased the detectability of ctDNA by SNP CMA. We successfully analyze 10/10 (100%) cfDNA samples by SNP CMA with cfDNA input amount as low as 30 ng. We also demonstrated the limit of detection for this approach with SCA and MNA detected at the lower limit of 20% and 10% of the sample, respectively.

The diagnostic accuracy of cfDNA for the determination of MNA status in advanced stage neuroblastoma was investigated since the beginning of the century (even before the concept of “liquid biopsy” emerged) using PCR techniques, with quantitative real-time PCR (qPCR) MNA detection reported in large cohorts of patients [4]. Here, we compared results between ddPCR and qPCR for MNA detection and found them both to reliably detect normal copy levels, *MYCN* gain, and MNA, with ddPCR demonstrating a superior sensitivity to detect CNAs.

The detectability of ctDNA from cfDNA is largely influenced by pre-analytical factors, as evaluated from data on SNVs, but to our knowledge was not assessed for CNAs. We evaluated the impact of pre-analytical procedures for plasma processing on the detectability of MNA in ddPCR. We found that EDTA blood collection tubes do not prevent the release of background cfDNA and so eliminate MNA detection. This was observed in conditions mimicking routine sample transfer in a hospital environment, both in artificial spiking experiments and in real-life patient’s samples scenarios.

There is great potential for ctDNA molecular diagnosis in neuroblastoma. In 2005, a case report by Combaret et al. highlighted the utility of *MYCN* evaluation in cfDNA, where provisional diagnosis was made but tumor biopsy was not possible. A 13-day-old infant presented with left adrenal tumor and elevated urinary catecholamine metabolites, however, thrombocytopenia and hypofibrinogenemia confounded the collection of tumor tissue. Evaluation of serum *MYCN* by qPCR revealed high-level MNA, and the child was assigned with high-intensity chemotherapy. Three weeks after a significant response, blood coagulation parameters had returned to normal and biopsy tissue was obtained, which showed greater than 50 *MYCN* copies by qPCR.

In our study, for patient ID Dx1 the initial biopsy suffered crush artefact and was difficult to interpret. In addition, biopsy at relapse was done neurosurgically (brain metastases) and so carried inherent clinical risks. On both occasions, cfDNA analysis demonstrated MNA and the ALK p.F1174L variant, and MIBG showed positive avidity. Thus, alternative approaches that overcome technically difficulties or high-risk surgical biopsy of the primary and/or recurrent tumors such as combining imaging and liquid biopsy can be very valuable.

Low-risk neuroblastoma represents 25% of cases, is classified in infants younger than 18 months, with either localized or metastatic (to the liver, skin, and bone marrow sites only) non-MNA disease. Clinically the disease is monitored without treatment as it usually spontaneously regress [24]. Molecular cytogenetics studies of low-risk tumors show hyperdiploidy (having extra chromosomes) rather than SCA. In this very young population, a surrogate marker to detect tumor-specific alterations in a non-invasive setting and avoiding tissue biopsy would be very desirable since accessibility of the tumor, morbidities associated with surgical risks and exposure to anesthetics (during MIBG imaging) can present key technical and practical challenges. While we show that localized neuroblastoma shed higher-than-normal cfDNA amounts, we were not able to analyze any of the four localized cases using SNP array due to low-input cfDNA levels. In addition, although several localized pediatric diseases were found to shed high ctDNA VAF in some cases (e.g., [25]), there was not enough to proceed with SNP CMA, given the assay sensitivity or LOD for identify low levels of MNA (10%) and SCA (20%). Next generation sequencing (NGS) is used routinely to detect DNA somatic variants when VAF ≥ 5% and might offer a better solution in this setting. However, NGS has its own limitations as benchmarking copy number variant callers tend to be challenging [26,27]. Currently, there are no molecular biomarkers for treatment response of high-risk disease. About half of patients with high-risk disease will harbor MNA and so MNA detection using ctDNA could present a potential biomarker for treatment response using either SNP array or ddPCR techniques. However, SNP CMA is limited in detecting MNA at clonal levels <10% and ctDNA input amounts (>30 ng) limit its use to only samples at initial presentation or recurrent disease. Using ddPCR with a detection limit >0.5% VAF for *MYCN* gain, we were able to monitor treatment response across the course of disease of six patients, testing either MNA (4 cases) or ALK variants (3 cases). We show that longitudinal testing can capture ctDNA dynamics and response to treatment. However, samples taken post neoadjuvant chemotherapy in four cases (Dx4, Dx3, Rec2, Dx1) showed only normal diploid *MYCN* levels, although imaging studies performed around the same time demonstrated MIBG avidity. A possible explanation to the difference between ctDNA and MIBG would be that ctDNA is detected in the context of proliferating cells only, while MIBG can detect live neuroblastoma cells, proliferating or static. MNA detected again at disease recurrence in three of these cases (75%). We conclude therefore that our test, having 0.5% VAF LOD, is not sensitive enough for early recurrence or minimal residual disease monitoring.

At the age of precision medicine, the clinical management of high-risk patients harboring hotspots *ALK* somatic variants is changing. ALK inhibitors are currently being investigated as upfront treatment for newly diagnosed patients with confirmed *ALK* mutated tumors. However, access to adequate core needle tissue to determine *ALK* somatic variants at initial diagnosis is challenging, with only 10% of samples reported to be successfully genotyped (either mutated or wild type), despite the prevalence of *ALK* somatic variants in this population being relatively high (14%). Our results support previous findings in small cohort cases of *ALK* variants in the circulation of patient with neuroblastoma. Thus, liquid biopsy for *ALK* variants could potentially overcome current hurdles in this clinical setting.

Several studies have demonstrated the suitability of serial *ALK* variants assessment for disease monitoring of ALK-positive non-small-cells lung cancer (NSCLC) [28]. Madsen and colleagues [29] reported a total clearance of *ALK* variants within two months after ALK inhibitors treatment initiation for patients achieving partial response. In our study *ALK* variants were detected in all three patients treated with Lorlatinib 6–8 weeks into the targeted treatment. Nonetheless, *ALK* variants’ trajectories reflected response to treatment and thus provide a surrogate biomarker to monitor targeted treatment efficacy in real-time. Sampling ctDNA at shorter intervals might provide earlier identification of treatment failure/success in neuroblastoma.

Unfortunately, it is becoming apparent that therapeutic resistance to ALK inhibition is inevitable for all current compounds. NGS studies demonstrated spatial and temporal evolution of neuroblastoma. Looking into a very narrow lens for characterizing only *MYCN* and *ALK* copy numbers, we found in this small study cohort that *ALK* gain of 4 copies in patient (ID Rec1) with available whole genome sequencing and high-depth targeted sequencing of synchronous tumor samples showed no changes to *ALK* copy number. However, using NGS for liquid biopsy could offer the opportunity to incorporate additional genomic findings about tumor response and result in earlier detection of somatic variants associated with resistance, and therefore increase treatment precision thereby improving patient survival and quality of life.

The strengths of our study include the use of ctDNA from neuroblastoma patients to identify SNVs and both DNA copy number amplification and SCA by ddPCR and SNP array, which involved optimization of sample collection conditions, novel use of the FFPE DNA restoration kit, evaluating the limit of detection of copy number abnormalities, and lower input ctDNA for SNP microarray analysis. Limitations of our study include the small cohort size, which partly reflects the rarity of neuroblastoma cases and partly the available cfDNA samples at the time of this study. Additional clinical sample are needed to more extensively assess the use of ctDNA, particularly for use of the CytoSNP-850K Beadchip assay. The small DNA fragment size of ctDNA and low input DNA amounts can pose challenges for the assay. In samples with poor quality assay metrics, segmental copy number abnormalities <5 Mb in size or at levels <20% of the sample may go undetected.

To summarize, neuroblastoma offers an attractive disease for ctDNA applications due to the very high tumor burden of most cases at initial presentation, and accordingly, very high ctDNA amounts. Neuroblastoma is one of few pediatric cancers in which biomarkers are routinely used for diagnosis, prognostication and therapeutic actions. Elevated urinary catecholamine metabolites and MIBG avidity, each evident in over 90% of patients, are routinely evaluated for provisional diagnosis and staging, and usually assessed prior to the biopsy procedure. This time window from the provisional diagnosis until the biopsy procedure, allows for cfDNA SCA and MNA profiling. Here, we show how SCA and MNA can be detected by SNP CMA within 1–2 weeks, which is the current diagnostic approach. We also demonstrate how ddPCR can be employed to provide orthogonal confirmation for MNA. With a turnaround time of 1–2 days for ddPCR, both could be achieved at a clinically relevant time frame while the biopsy procedure is been planned. Therefore, reducing core needle tissues could be considered in the context of provisional diagnosis and positive ctDNA-SCA or -MNA findings.

## 5. Conclusions

Neuroblastoma has been shown to shed important amounts of ctDNA in particular in metastatic cases and high-risk disease and thus represents a relevant model for ctDNA analysis. We show that ctDNA can be used as a surrogate marker for molecular diagnosis of SCA, MNA, and *ALK* variants. We present strategies for optimizing plasma collection and ctDNA analysis to reduce input requirements and ensure the accuracy and reproducibility of SCA and/or MNA liquid biopsy test. Finally, we suggest a framework for future implementation of liquid biopsy for neuroblastoma diagnosis to reduce tissue requirements currently embedded in the management of neuroblastoma.

## Figures and Tables

**Figure 1 cancers-13-03365-f001:**
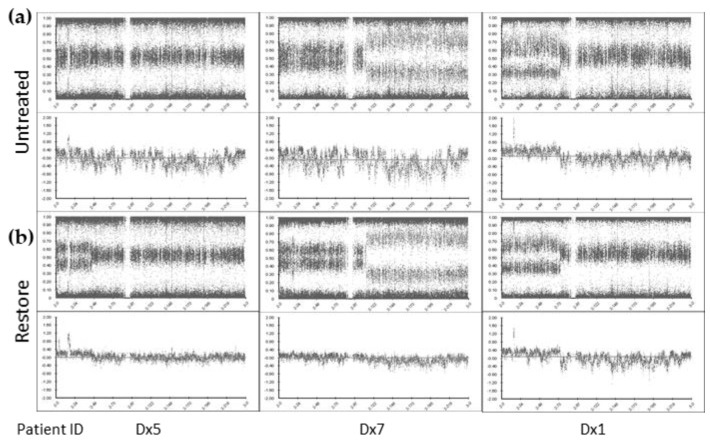
cfDNA treatment enables clearer identification of SCA and MNA by SNP CMA. Chromosome 2 view of SNP CMA data of (**a**) untreated cfDNA and (**b**) cfDNA treated with the Infinium FFPE DNA Restoration kit (Restore). BlueFuse Multi v4.5 B-allele frequency (top) and log R ratio (bottom) profiles presented.

**Figure 2 cancers-13-03365-f002:**
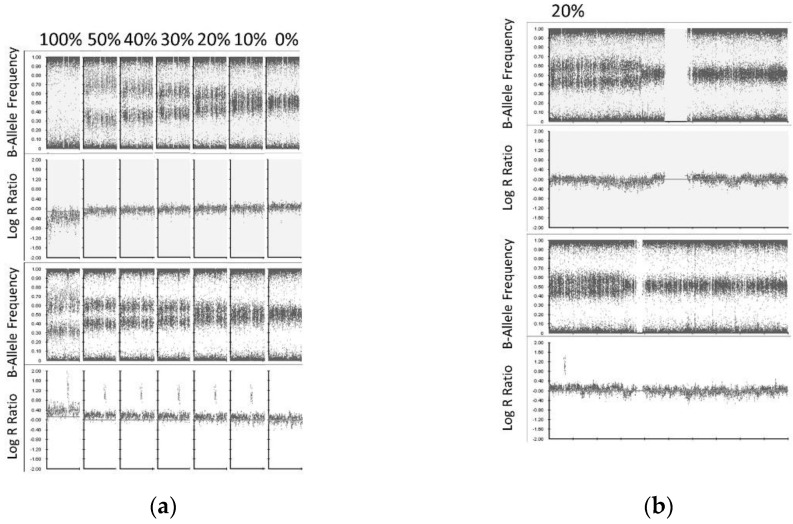
ctDNA SNP array limit of detection for SCA and MNA. (**a**) cfDNA was titrated into matched [fragmented] constitutional DNA according to the level indicated for chromosome 1p36.33p36.12 loss (top) and 2p25.3p24.1 gain (bottom) presented. SCA were detected at a level of 20% (indicated by B-allele frequency), and MNA at a level of 10% (indicated by log R ratio). (**b**) Whole chromosome view (chromosome 1 top, chromosome 2 bottom) showing 1p and 2p SCA with MNA at level of 20% of the sample.

**Figure 3 cancers-13-03365-f003:**
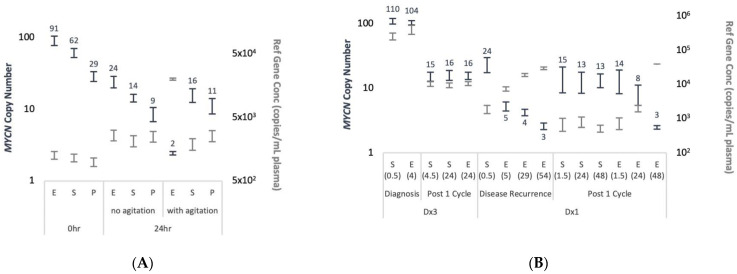
Blood collection tube type and time till processing affect the levels of cfDNA MNA detection. (**A**) Normal blood samples collected in EDTA (E), Streck (S), and PAXgene (P) tubes, spiked immediately with Kelly cell line DNA, and processed to plasma at the indicated time post venipuncture with or without agitation. (**B**) Matched samples collected from patients diagnosed with MNA-neuroblastoma at initial presentation prior to treatment (diagnosis) or at disease recurrence, and at after 1 cycle of chemotherapy. Blood was collected in EDTA (E) and Streck (S) tubes and processed at the indicated time post venipuncture (in brackets) after incubation at room temperature without agitation. Following cfDNA extraction from plasma *MYCN* copy number (navy, left Y axis, values showed on outside end) and *THNSL2* concentrations (grey, right Y axis) assessed by duplex ddPCR and presented with 95% CI.

**Figure 4 cancers-13-03365-f004:**
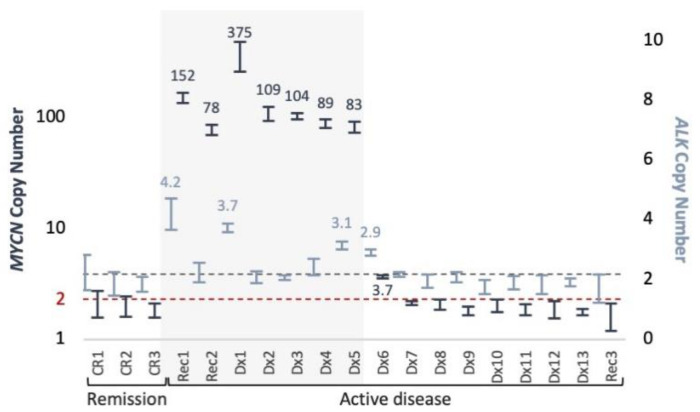
*MYCN* and *ALK* copy number in plasma samples. ddPCR triplex copy number assessment of *MYCN* (navy, left Y axis), and *ALK* (light blue, right Y axis), presented with 95% CI. Elevated levels indicated. Background shading indicates MNA cases according to diagnostic information of tumor samples. Dashed lines indicate normal diploid copy levels.

**Figure 5 cancers-13-03365-f005:**
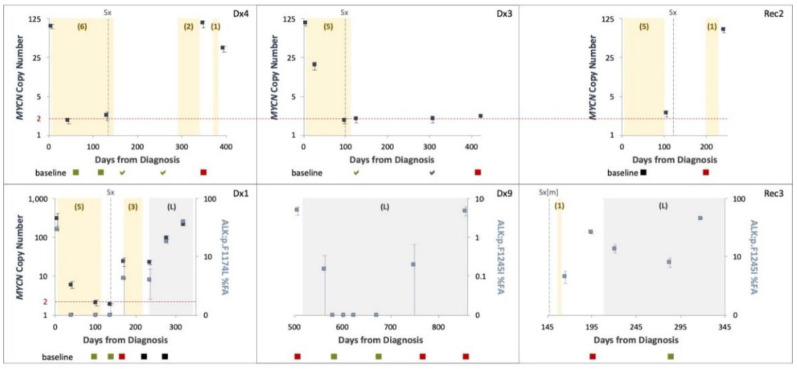
ctDNA levels and patterns throughout the clinical course of disease. *MYCN* copy number (navy, left Y axis) and *ALK* variants fractional abundance (FA) (light blue, right Y axis) assessed from serial plasma samples using ddPCR presented with 95% CI. Chemotherapy (number of cycles in brackets) and Lorlatinib (L) therapy are indicated by shaded yellow and grey areas, respectively. Surgical resection of the primary tumor (Sx), or lung metastasis (Sx[m]) are indicated. Dashed horizontal lines indicate normal *MYCN* diploid copy levels. MIBG imaging presented at the bottom of each graph performed at indicated time points: <- evidence of disease, a- no evidence of disease; green–disease regression, black–stable disease, red–disease progression.

**Figure 6 cancers-13-03365-f006:**
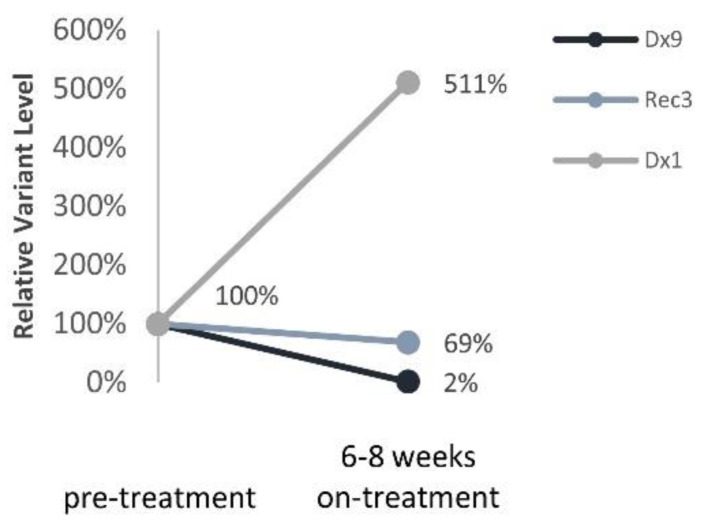
*ALK* ctDNA levels in response to ALK inhibitor treatment. Relative ALK variants FA change pre- and 6–8 weeks on- treatment with Lorlatinib.

## Data Availability

The data presented in this study are available on request from the corresponding author.

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
