# Peer review of "Neuroblastoma Molecular Risk-Stratification of DNA Copy Number and ALK Genotyping via Cell-Free Circulating Tumor DNA Profiling"

_cancers, 2021, doi:10.3390/cancers13133365_

Round 1

Reviewer 1 Report

Overall, it is a good study. The aims are simple and clear, while the design is correct. The results are interesting and discussion adequately recognizes both possibilities and limitation of the proposed approach.

The reported cfDNA concentrations are extremely high (as clearly stated by the authors), but in line with the previous reports. In terms of reporting, the distribution seems highly skewed (“21–9,980 ng/mL of plasma, mean 1,254 ng/mL, median 172 ng/mL”), so median in addition to / instead of the mean would be more informative in “mean 2,426 ng/mL for metastatic disease versus 54 ng/mL for localized disease”.

I really liked the concept of employing FFPE DNA Restoration kit to improve the quality of cfDNA. I have not seen this approach before, but it is logical (with fragmentation being the problem in both cases) and it seems to work.

Figure 3 could be improved for readability – choosing more contrasting colors and presenting the values in separate lanes (as in Fig.4) would be helpful, as they sometimes overlap. The same for Figure 5.

There are some punctuation errors and some problems with citations, so a careful check would be helpful.

Reviewer 2 Report

Smadar Kahana-Edwin et.al. presented how liquid biopsy could provide neuroblastoma molecular risk-stratification. This is very important in cases that tumor tissue is unavailable. However I have some major issues to point out:

  1. In this study only sixteen patients were enrolled. This is a small cohort. Authors should analyze a bigger amount of samples.
  2. Page 3, lines 131-140 : in the case of Streck tubes the second centrifugation of the plasma supernatant (15,500 x g for 10 mins) is performed before storing of plasma, while in the case of EDTA tubes is performed after the storing of sample. Since you are comparing the performance of these two collection tubes you should follow  the same protocol.
  3. Authors refer that Cell free circulating DNA was extracted from 0.25-3 mL. Why authors did not use a constant volume of sample? In cases of starting volume of plasma lower than 3 why was the plasma diluted with PBS? 
  4. Authors refer that 150 ng DNA was used so as to define LOD of the assay. However in some cases of clinical samples less than 150ng were used (30-150). Ιn such cases, how authors ensure that the method is of high sensitivity? 
  5. Page 8, line 351: why was the spiking performed in the whole blood and not directly to the plasma sample? My main observation refers to the fact that plasma isolation can be performed directly after blood collection, while the storing of plasma in -80, is critical for the preservation of ctDNA.

Round 2

Reviewer 2 Report

I believe that authors respond to my comments.